# The Effects of Short-Term Changes in Sodium Intake on Plasma Marinobufagenin Levels in Patients with Primary Salt-Sensitive and Salt-Insensitive Hypertension

**DOI:** 10.3390/nu13051502

**Published:** 2021-04-29

**Authors:** Katarzyna Łabno-Kirszniok, Agata Kujawa-Szewieczek, Andrzej Wiecek, Grzegorz Piecha

**Affiliations:** Department of Nephrology, Transplantation and Internal Medicine, Medical University of Silesia, Francuska 20/24, 40-027 Katowice, Poland; katarzyna.labno.kirszniok@gmail.com (K.Ł.-K.); agata.szewieczek@gmail.com (A.K.-S.); awiecek@sum.edu.pl (A.W.)

**Keywords:** marinobufagenin, salt sensitivity, salt-induced hypertension, NT-proANP

## Abstract

Increased marinobufagenin (MBG) synthesis has been suggested in response to high dietary salt intake. The aim of this study was to determine the effects of short-term changes in sodium intake on plasma MBG levels in patients with primary salt-sensitive and salt-insensitive hypertension. In total, 51 patients with primary hypertension were evaluated during acute sodium restriction and sodium loading. Plasma or serum concentrations of MBG, natriuretic pro-peptides, aldosterone, sodium, potassium, as well as hematocrit (Hct) value, plasma renin activity (PRA) and urinary sodium and potassium excretion were measured. Ambulatory blood pressure monitoring (ABPM) and echocardiography were performed at baseline. In salt-sensitive patients with primary hypertension plasma MBG correlated positively with diastolic blood pressure (ABPM) and serum NT-proANP concentration at baseline and with serum NT-proANP concentration after dietary sodium restriction. In this subgroup plasma MBG concentration decreased during sodium restriction, and a parallel increase of PRA was observed. Acute salt loading further decreased plasma MBG concentration in salt-sensitive subjects in contrast to salt insensitive patients. No correlation was found between plasma MBG concentration and left ventricular mass index. In conclusion, in salt-sensitive hypertensive patients plasma MBG concentration correlates with 24-h diastolic blood pressure and dietary sodium restriction reduces plasma MBG levels. Decreased MBG secretion in response to acute salt loading may play an important role in the pathogenesis of salt sensitivity.

## 1. Introduction

High dietary salt intake promotes the development of arterial hypertension (AH) while dietary salt restriction is associated with the decrease of blood pressure (BP) in both hypertensive and normotensive patients [1,2,3,4,5]. The observed heterogeneity and individual blood pressure response to short-term changes in sodium intake allowed for the categorization of patients as salt-sensitive or salt-insensitive [6]. The salt-sensitivity of arterial blood pressure is estimated at 51% of the hypertensive population and 26% of normotensive individuals [7] and occurs more frequently in women, the Black population and in older patients [7,8,9]. It is supposed that a high-salt diet is a risk factor for cardiovascular disease and organ damage [10,11] and increases the risk of death, independently of mean blood pressure [12].

The pathogenesis of salt sensitivity of blood pressure is not entirely understood. In-sufficiently suppressed renin–angiotensin system (RAS), decreased atrial natriuretic pep-tide (ANP), nitric oxide (NO), bradykinin and kallikrein secretion, enhanced sympathetic activity [13,14], some genetic predispositions [15] and immune mechanisms [16] have been proposed to contribute to the increased sodium and water retention in salt-sensitive individuals. It has been postulated for some time that endogenous hormones are stimulated by high-salt dietary intake and may increase natriuresis by inhibiting renal tubular Na/K-ATPase (NKA), a heterodimer consisting of alpha, beta and gamma subunits. Over half a century ago, Dahl et al. formulated a hypothesis of a circulating “humoral factor” that may induce salt-sensitive arterial hypertension [17,18]. These “humoral factors”, are endogenous cardiotonic steroids (CTS) similar to aglycone bufadienolides (such as marinobufagenin (MBG)) and cardienolides (such as ouabain) [19]. A number of α, β and γ NKA subunit isoforms have been described, which are expressed in a tissue-dependent manner and differ in terms of sensitivity to different CTS between species [20,21]. As was previously pointed out, the cardienolides have a predilection for the α2 and α3 isoforms which are expressed in vascular smooth muscle cells (VSMC), skeletal muscle, heart, central and peripheral nervous system. Additionally, the bufadienolides may inhibit also the α1 isoform which is expressed in most tissues and organs [20].

Increased circulating levels of cardiotonic steroids have been proposed as a compensatory mechanism for natriuresis and vascular tone in volume-expanded conditions. Marinobufagenin initiates natriuresis by NKA inhibition directly in renal tubules through reduced renal sodium reabsorption and increased sodium excretion. Moreover, by inhibiting the NKA in vascular smooth muscle cells and vasoconstriction promotes natriuresis in a pressure-induced mechanism [20,22]. Additionally, via an alternative NKA- dependent “signaling pathway” MBG promotes fibrosis and hypertrophy in the arterial wall, heart and kidney [20].

Data from experimental studies support the hypothesis that both acute and long-term plasma volume expansion due to intravenous saline infusion [23] or a high-salt diet [24] lead to increased synthesis of MBG. Similarly, the same effect of high salt intake was noted in a salt-sensitive animal model [25,26]. It should be emphasized that most of these observations came from experimental studies and human data are scarce and include only small groups of patients. 

The main aim of this study was to determine the effects of short-term changes in sodium intake on plasma MBG levels in patients with primary salt sensitive and salt-insensitive hypertension.

## 2. Materials and Methods

### 2.1. Study Group

Fifty-one adult patients with primary hypertension were enrolled in this prospective study. The exclusion criteria were: secondary arterial hypertension, systolic blood pressure (SBP) > 170 mmHg and/or diastolic blood pressure (DBP) > 100 mmHg despite the use of antihypertensive drugs, episodes of pulmonary edema in the last 6 months, a weight gain of more than 2.5 kg after withdrawal of diuretics, impaired kidney function at the time of the study with estimated glomerular filtration rate (eGFR) ≤ 60 mL/min/1.73 m^2^ (calculated according to the MDRD formula), diagnosis of sever liver disease or malignancy and expected non-compliance. Diuretic therapy was discontinued in all treated patients at least 3 days before this study began. 

This prospective study was conducted for 4 days and four observation points were evaluated: at baseline (stage A), after a 3-day low sodium diet (≤20 mmol Na/day) (stage B), after acute intravenous sodium chloride load (310 mmol sodium, 2 L of 0.9% sodium chloride infused over 4 h) (stage C) and 24 h after the salt loading (stage D). The study protocol was accepted by the Bioethics Committee of the Medical University of Silesia in Katowice, Poland (KNW/0022/KB1/42/I/13) and all participants provided written informed consent.

### 2.2. Laboratory Measurements

Blood samples were collected from each patient in the morning after >8 h of fasting. After collection, blood samples were immediately centrifuged, serum and plasma were aliquoted in 1 mL test tubes and processed immediately or frozen in −70 °C until analysis.

Plasma concentration of MBG, serum concentration of *n*-terminal atrial natriuretic pro-peptide (NT-proANP), *n*-terminal B-type natriuretic pro-peptide (NT-proBNP), sodium, potassium as well as hematocrit (Hct) values were assessed at each of the 4 stages of this study. Serum concentrations of aldosterone and plasma renin activity (PRA) were assessed at stage 1–3 of this study. Urinary sodium and potassium excretion was measured in 24-h urine collection at the beginning (stage A), during dietary sodium restriction (stage B) and during acute salt loading (stage C–D).

Plasma MBG concentration was measured following solid-phase extraction with C-18 columns (Waters, Cambridge, MA, USA), as described previously [27]. Samples were placed in assigned pairs (salt-sensitive and salt-insensitive) on the same plate. Measurements were repeated three times with sample pairs rearranged on different plates between repeated measurements. An average of these three measurements was used as case value for analysis.

An enzyme-linked immunosorbent assay (ELISA) using commercially available kits was used for measurements of NT-proANP serum concentration (DuoSet Elisa, R&D Systems, Minneapolis, MN, USA). Serum NT-proBNP levels were assessed by the electrochemiluminescence method (ECLIA), using commercially available kits in Elecsys 2010 analyzer (Roche Diagnostics, Basel, Switzerland). PRA and aldosterone serum concentrations were measured by radioimmunoassay with commercially available kits (Beckman Coulter, Immunotech s.r.o., Prague, Czech Republic and ZenTech, RIAZENco, Angleur, Belgium, respectively). Urine and serum concentrations of sodium and potassium were assessed by ion-selective electrode method (SYNCHRON Systems, Beckman Coulter, Brea, CA, USA) and other blood assays were performed using standard methodology.

### 2.3. Blood Pressure Measurements

Patients with BP values ≥140/90 mmHg or those who received antihypertensive medication were diagnosed as hypertensive and the diagnosis of arterial hypertension was made prior to admission to our department. During our observation period, blood pressure was measured 2–3 times on the left arm, after more than a 5-min rest in sitting position and such blood pressure measurements were performed at baseline, during sodium restriction and sodium load. The mean arterial blood pressure (MAP) was calculated for each period of sodium intake. The salt sensitivity of arterial blood pressure was diagnosed in those patients whose MAP on a low-sodium diet was at least 10 mmHg lower than the MAP value after sodium load [7]. Patients with a lower decrease or with blood pressure increase during a low-sodium diet were classified as salt-insensitive. A 24-hour ambulatory blood pressure monitoring (ABPM) was performed at baseline, using an oscillometric method (TM-2430 device).

### 2.4. Echocardiography

Echocardiographic measurements were performed at baseline using the Toshiba Aplio 400 Diagnostic Ultrasound System (Toshiba, Toshiba Medical System Corporation, Utsunomiya, Japan). M-mode and two-dimensional measurements were applied as recommended by the American Society of Echocardiography [28], including left ventricular end-diastolic and end-systolic diameters, intraventricular septum, and posterior wall end-diastolic thickness. Left ventricular mass (LVM) was calculated according to the Devereux formula [29]. LVM was indexed for BSA (LVMI). Next, relative wall thickness (RWT) was calculated according to the formula proposed by Lang et al. [30]. Left ventricular hypertrophy was diagnosed following the European Society of Hypertension (ESH) and the European Society of Cardiology (ESC) recommendations [31] and the chamber quantification was performed according to references [30,32].

Moreover, all patients underwent an ocular fundus examination with a direct ophthalmoscopy.

### 2.5. Statistical Analysis

Statistical analysis was performed using the Statistica 12.0 PL software (StatSoft Pol-ska, Cracow, Poland). The level of statistical significance was set at a *p* value below 0.05. The Shapiro–Wilk’s test was used to assess the distribution of variables and results are presented as median values with interquartile range. The differences between groups were assessed using Mann–Whitney U-test or Student’s t-test (for variables with non-normal or normal distribution, respectively). To compare variables in two or more groups the Kruskal–Wallis test (for independent variables) and Wilcoxon or Friedman tests (for dependent variables) were used. Categorical variables were compared using either Fisher′s exact test or χ2 tests with Yates correction. The correlation between variables was assessed with Spearman’s rank correlation coefficient and was calculated for determinants simultaneously assessed in both patient groups.

## 3. Results

### 3.1. Study Groups Characteristics

Fifty-one adult patients with primary hypertension (34 females and 17 males) were enrolled in this prospective study. The median age was 52 years (42.5; 59.5). All participants completed the study and 27 were diagnosed as salt-sensitive (subgroup 1; 20 females and 7 males) and 24 as salt-insensitive (subgroup 2; 14 females and 10 males). From those, 18 patients were salt-resistant (<5 mmHg decrease or an increase in MAP) and 6 had indeterminate BP change of −5 to −9 mmHg. There was no significant difference between these groups in age, gender, body mass index (BMI) and eGFR. Notably, there was no significant difference at baseline in ABPM and in echocardiographic parameters between salt-sensitive and salt-insensitive individuals. The percentage of patients with hypertension-related target organ damage was similar in both groups. Concentric hypertrophy was the most common abnormal LV geometric pattern in both subgroups (66.7% vs. 45.8% in subgroups 1 and 2, respectively; *p* = NS). Furthermore, there was no difference in baseline plasma MBG levels, serum concentrations of NT-proANP, NT-proBNP, aldosterone and potassium, as well as hematocrit value and PRA between both subgroups. Neither urinary sodium nor potassium excretion differed significantly between analyzed subgroups. However, salt-sensitive patients showed significantly higher serum sodium concentration median values than salt-insensitive individuals at baseline (140.3 (139.0–141.0) vs. 139.0 (138.0–139.9) mmol/L respectively, *p* < 0.05). The characteristics of the study groups are shown in Table 1.

Both systolic and diastolic office blood pressure decreased significantly after salt restriction in the salt-sensitive group, but not in the salt-insensitive group (Figure 1).

### 3.2. Short-Term Changes in Sodium Intake and Plasma MBG Levels

After both sodium dietary restriction (stage B) and intravenous sodium load (stage C), no significant differences were found between the study groups in the plasma MBG concentration. However, crucial changes in plasma MBG concentration were noted at subsequent stages of observation only in the salt-sensitive group. Dietary sodium restriction was associated with a significant decrease in plasma MBG concentration in this subgroup (0.300 (0.185; 0.446) vs. 0.342 (0.243; 0.540) nmol/L, stage B vs. A, respectively, *p* < 0.01). Furthermore, plasma MBG concentration measured 24 h after salt loading (stage D) declined significantly in this subgroup compared to the value immediately after the 0.9% NaCl infusion (stage C) (0.223 (0.124; 0.328) vs. 0.290 (0.213; 0.434) nmol/L, respectively, *p* < 0.001) and to the value at baseline (stage A) (0.223 (0.124; 0.328) vs. 0.342 (0.243; 0.540) nmol/L, respectively, *p* < 0.01) (Figure 2a). On the contrary, no significant effect of short-term changes in sodium intake on plasma MBG levels was observed in the salt-insensitive group.

### 3.3. Short-Term Changes in Sodium Intake and NT-proANP, Aldosterone, PRA, Sodium and Potassium Levels

In both groups, a significant decrease in serum NT-proANP concentration after 3 days of low sodium diet was observed, followed by a significant increase immediately after salt-loading and another decrease at stage D (Figure 2b). There was no difference in serum NT-proANP concentration between the groups. During the follow-up period assessment typical and similar changes in aldosterone and PRA levels were noted in both study groups (Figure 2c,d). The serum levels of aldosterone and PRA levels were comparable in both study groups at subsequent stages of the observation. Similarly, the analyzed groups did not differ significantly with regard to values of hematocrit, sodium and potassium serum concentrations and sodium and potassium urinary excretion measured at subsequent stages of the sodium restriction and sodium load (Figure 2e–i). Notably, the evaluation of hematocrit values after acute salt loading revealed, that salt-insensitive patients were characterized by a slight increase in hematocrit values between period C and D (38.45% (37.78; 40.33) vs. 38.50% (36.40; 40.53), respectively, *p* < 0.05). Conversely, in salt-sensitive individuals, hematocrit values 24 h after sodium load were lower than immediately after the infusion of sodium chloride (38.30% (37.40; 39.80) vs. 39.40% (38.20; 40.45), respectively, *p* < 0.05).

### 3.4. Plasma Marinobufagenin Levels at Baseline and LV Quantification

There was no association between the type of left ventricular geometry, the grade of LVH and plasma MBG levels either in all hypertensive patients, or in both subgroups analyzed separately (data not shown).

### 3.5. Correlation between MBG and Analyzed Parameters

There was a significant positive correlation between plasma MBG and systolic blood pressure at baseline in the salt-sensitive group (r = 0.45; *p* < 0.02) but not in the salt-insensitive group (r = 0.005; *p* = NS). At later stages this correlation did not reach statistical significance.

A significant positive correlation was found between plasma MBG and serum NT-proANP concentrations in both analyzed groups at the baseline. Of note, this association was more pronounced in salt-sensitive subjects and, as opposed to salt-insensitive patients, was still present after low sodium diet (baseline: r = 0.50, *p* < 0.01 and r = 0.42, *p* < 0.05, respectively; follow-up: r = 0.58, *p* < 0.001 and r = 0.11, *p* > 0.05, respectively) (Figure 3). Moreover, we revealed a significant positive correlation between changes in plasma MBG concentration and changes in PRA (calculated as a difference of values between subsequent points A and B of this study) during the sodium restriction period in the salt-sensitive group (r = 0.42, *p* < 0.05) but not in salt-insensitive patients (Figure 4). There was no significant correlation between the change in NT-proANP or change in aldosterone with change in MBG in either group at the same time (data not shown). No significant correlation was found between plasma MBG concentration and serum sodium or potassium concentration or urinary sodium excretion at any point in the study. There was also no significant correlation between changes in these parameters. We found a significant correlation between plasma MBG concentration and serum aldosterone concentration at baseline in salt-insensitive individuals (r = 0.41, *p* < 0.05) but not in salt-sensitive patients. No significant correlation was found between plasma MBG levels and left ventricular mass index in patients with primary hypertension. At baseline, plasma MBG concentration correlated with a 24-h diastolic blood pressure only in salt-sensitive individuals (r = 0.57, *p* < 0.01).

## 4. Discussion

This study is so far one of few publications concerning the role of marinobufagenin in the pathogenesis of salt-sensitivity of blood pressure. Results obtained in our study have shown that in salt-sensitive hypertensive patients, plasma MBG concentration correlates positively with 24-h diastolic blood pressure and NT-proANP levels. Additionally, plasma concentration of MBG declined during 3 days of low sodium intake, paralleled by an increase in PRA in this subgroup. Acute salt loading further significantly decreased plasma MBG concentration in salt-sensitive hypertensive subjects compared to salt-insensitive ones. We did not find significant association between plasma MBG levels and left ventricular mass index in patients with primary hypertension, regardless of their salt-sensitivity.

There are several possible patomechanisms linking endogenous cardiotonic steroids with salt-induced hypertension and organ damage [20]. It has been postulated that endogenous ouabain (EO), which is produced and released in the brain in response to the initial increase in cerebrospinal fluid sodium, precedes the increases in BP [33]. Ouabain, as a neurohormone, activates the local RAS and SNS [34], leads to increased synthesis of angiotensin II and finally to the release of MBG in the adrenal cortex [35]. Experimental studies revealed that both acute [23,36] and chronic [24] sodium chloride load may stimulate the synthesis of marinobufagenin. Similar observations were made in humans; however, limited data are available and they include small groups of patients with normal or moderately elevated blood pressure [37,38,39] or with resistant hypertension [40]. Moreover, it was postulated that increased MBG secretion after high sodium load is an attempt to compensate for impaired pressure-natriuresis in the animal model of salt-sensitivity [25]. So far, no studies have been conducted to determine the effects of short-term changes in sodium intake on plasma MBG levels in patients with primary hypertension depending on salt sensitivity. In the present study, significant changes in plasma MBG concentration induced by different sodium intake were noted only in salt-sensitive hypertensive patients but not in salt-insensitive ones. Short-term dietary sodium restriction was associated with a significant decrease in MBG plasma levels, despite the fact that we did not find a significant correlation between plasma MBG concertation and urinary sodium excretion in this subgroup. Our observation is indirectly in line with a previously published small study in 11 patients with moderately elevated systolic BP (139 ± 2/83 ± 2 mmHg) [37]. Although the authors showed that plasma MBG levels did not differ between various dietary periods, urinary excretion of MBG was lower during the low-sodium condition. Nevertheless, a positive correlation between urinary sodium excretion and plasma MBG concentration during the low-sodium diet was not confirmed, neither in the above mentioned, nor in other previously published studies [37,38].

Interestingly, there was no significant increase in the plasma MBG level in response to acute sodium chloride load in our cohort, in contrast to experimental studies. Intraperitoneal NaCl load has been shown to increase both plasma concentration and urine MBG excretion in Dahl salt-sensitive rats [25]. Available data in humans show an increase in plasma MBG concentration after several days of increased salt intake. Anderson et al. [38] showed, that the change from a lower (0.7 mmol/kg/day for 6 days) to a higher (4 mmol/kg/day for 6 days) dietary sodium intake in healthy, normotensive women (*n* = 28) was associated with a significant sustained increase in urinary MBG excretion, which in turn positively correlated with the fractional sodium excretion. Parallel to this observation, other authors noted that high-salt intake (for 4 weeks) induced a small but significant elevation in plasma MBG concentration, whereas urinary MBG excretion has not changed [39]. Only one study, performed in 34 patients with resistant hypertension, showed that acute salt loading for 60 min via intravenous infusion of 1 L of 0.9% NaCl resulted in the increase of plasma MBG concentration immediately after the salt loading [40]. In contrast to our observation all patients received a thiazide diuretic and salt loading resulted in a substantial BP rise. Experimental data indicate that sodium-induced changes in urinary MBG excretion may be several times higher than those observed in plasma [24,41], which was also confirmed by the above-mentioned study in humans [37]. As was previously pointed out by other investigators, the overall plasma MBG level depends upon the adrenal marinobufagenin secretion, diurnal variations typical for steroid hormones and the supposed kidney role in local production, and metabolism and storage of MBG [25,37]. Furthermore, in all previously reported studies except one [40] the condition defined as high salt-intake was induced through increased salt ingestion for 6 days or for 4 weeks, whereas in our cohort we used intravenous sodium chloride load, in the amount of 2 L infused over 4 h. Notably, when we evaluated hematocrit values we found that in salt-sensitive individuals, hematocrit values were lower 24 h after sodium load than immediately after the infusion of sodium chloride, where the opposite occurs in salt-insensitive patients. One can assume that this observation in the salt-sensitive sub-group was a consequence of sodium and water retention. Although salt-sensitive patients were characterized by slightly higher sodium concentrations in the serum and lower urine excretion at 24 h after sodium load, the difference between both analyzed groups did not reach statistical significance. 

The RAA system is one of the main modulators of the pressure natriuresis, which maintains a relatively constant mean arterial pressure despite significant changes in dietary sodium intake. It has been suggested that the salt-sensitivity of BP may be related to in-appropriate activation of the RAA system [7,42,43]. In the present study we observed typical changes in PRA and aldosterone levels in subsequent periods of sodium restriction and sodium load. However, in contrast to some studies [7,42,44], but in line with Gerdts et al. [45], no difference in plasma renin activity and serum aldosterone concentration was found between salt-sensitive and salt-insensitive subjects. Previously, it was demonstrated that plasma MBG levels were significantly higher in patients with primary aldosteronism compared to those with essential hypertension [46]. In a recently published case-controlled analysis including haemodialysis patients and matched controls we have shown a positive correlation between plasma MBG and serum aldosterone concentrations in the pooled population [27]. In the present study, plasma MBG levels correlated with the aldosterone only in salt-insensitive individuals at baseline and a lower decline in plasma MBG concentration was associated with more pronounced increase in PRA levels during the low sodium diet in salt-sensitive subjects. This observation may suggest a more pronounced effect of MBG in the short-term response to salt intake in salt-sensitive patients with hypertension. We should stress that there are different criteria for sodium sensitivity in the literature and authors used various methods measuring BP. Moreover, we could not withdraw some hypotensive drugs (except for diuretics) in all patients, so we cannot exclude their impact on PRA and serum aldosterone concentration.

Previous experimental studies revealed that ANP modulates MBG- induced NKA inhibition. ANP sensitizes the sodium pump in the kidney to the inhibitory effect of MBG and enhanced natriuresis, whereas in aorta ANP attenuates the MBG effect and promotes vasodilatation [47]. Dahl-salt-sensitive compared to normotensive Sprague-Dawley rats were characterized by a similar response in MBG secretion but a diminished response in ANP production due to NaCl loading [48]. This association was age-dependent and more pronounced in older animals [49]. Therefore, experimental evidence suggests that impaired secretion of ANP in salt-sensitive individuals may decrease natriuretic properties of MBG and intensify its vasoconstrictive effect. Given the considerable variability between species in terms of α1 NKA sensitivity to CTS experimental data should be carefully interpreted. Human data are limited, and demonstrated MBG and ANP interaction under normal sodium load condition in patients with primary hypertension [50,51] or with heart failure [52,53]. Results obtained in our study have shown a positive correlation between plasma MBG and NT-proANP concentrations in both subgroups at baseline. However, this correlation was present after sodium dietary restriction only in salt-sensitive patients.

Patients diagnosed in our study as salt-sensitive tended to have slightly higher SBP and DBP, a lower decline of BP at night (in ABPM) and an increased incidence rate of hypertensive retinopathy and LVH. However, these differences were not significant when we compared this subgroup to salt-insensitive subjects. When considering the possible association between MBG and BP during high-sodium intake, a positive correlation was observed in some studies, but exclusively in men (only for plasma MBG) [39] and after long-term high sodium ingestion [37,39]. On the other hand, in the first few days of NaCl loading the increased in MBG excretion was inversely related to SBP in middle-aged or older women and urine MBG declined with age [38]. Finally Strauss et al. [54] recently reported that in a group of 331 young and apparently healthy participants on a habitual diet a positive association of SBP with MBG/Na + ratio was proven only in black women. These observations suggest that the interaction between MBG and salt-sensitivity of blood pressure is modified by age, race and gender and is determined by the duration of high-sodium load. A limitation of our study is the lack of conducted ABPM during sub-sequent periods of the study, however we showed a significant correlation between plasma MBG concertation and 24-h diastolic blood pressure, only present in salt-sensitive individuals. This association was noted at baseline, under uncontrolled “normal” sodium dietary ingestion, which usually exceeds at least twice the recommended standards [55].The mean dietary sodium intake in our cohort at the beginning of the study was estimated based on urinary sodium excretion at approximately 122 mmol/day, which is higher than recommended by the WHO but lower than the average sodium intake in Europe [56]. 

Experimental data demonstrated that enhanced MBG synthesis during the high salt diet contributed to tissue remodeling in the heart, irrespective of blood pressure [24]. Evidence from a recently-published study in humans indicated a significant positive correlation between plasma MBG concentration and LVMI in young adults with excessively high MBG levels [54], especially in obese individuals [57]. Pierdomenico et.al. [58], evaluating the association between endogenous ouabain and LV geometry in patients with recently diagnosed essential hypertension, revealed that EO levels tended to be higher in subjects with concentric remodeling than in those with eccentric non-dilated hypertrophy. Of note is the fact that there are no human studies investigating the interaction between MBG and structural alterations in myocardium. Echocardiographic measurements performed at baseline in our cohort revealed that the prevalence of abnormal LV geometry was similar in both subgroups, and the mean recognized geometry type was concentric hypertrophy. Regardless of the type of left ventricular geometry and the grade of LVH plasma marinobufagenin levels tended to be similar in all analyzed subgroups and we did not find a significant correlation between the plasma MBG level and left ventricular mass index in our cohort. 

The present study has some limitations. The small size of our cohort determined the lack of subanalysis stratified by gender and age, or multivariate analysis of factors independently associated with salt-sensitivity of BP. We did not measure plasma EO and urinary MBG excretion and such observation, especially in the first day of sodium load, is warranted. Another limitation is the lack of prolonged 24-h ambulatory blood pressure monitoring with subsequential MBG measurements in the plasma and in the urine in all of study periods. Such observation would allow us better understand the effect of short-term changes in sodium intake. As we mentioned above, we cannot exclude the impact of used hypotensive drugs on PRA and aldosterone concentration. However, it is important to notice that neither analyzed group differed at the baseline in terms of demographics, co-morbidity, kidney function and prevalence of hypertension-related target organ damage.

## 5. Conclusions

In salt-sensitive hypertensive patients, plasma MBG concentration correlates positively with 24-h diastolic blood pressure, and dietary sodium restriction reduces plasma MBG level. The hypotensive effect of reduced dietary sodium may be due to either a decline in adrenal MBG synthesis and release or an increase in MBG elimination in salt-sensitive subjects. Decreased MBG availability in response to acute salt loading resulting in sodium retention may play an important role in the pathogenesis of salt sensitivity. Further investigation is warranted to fully elucidate this hypothesis.

## Figures and Tables

**Figure 1 nutrients-13-01502-f001:**
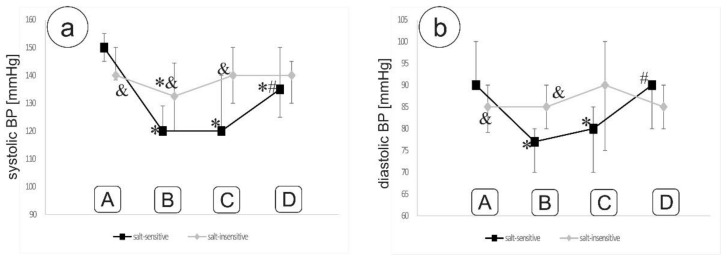
Systolic (**a**) and diastolic (**b**) office blood pressure throughout the study. * *p* < 0,05 vs. stage A; # *p* < 0,05 vs. stage C; and & *p* < 0.05 between groups. Description of stages A–D of the study is shown in Figure 2.

**Figure 2 nutrients-13-01502-f002:**
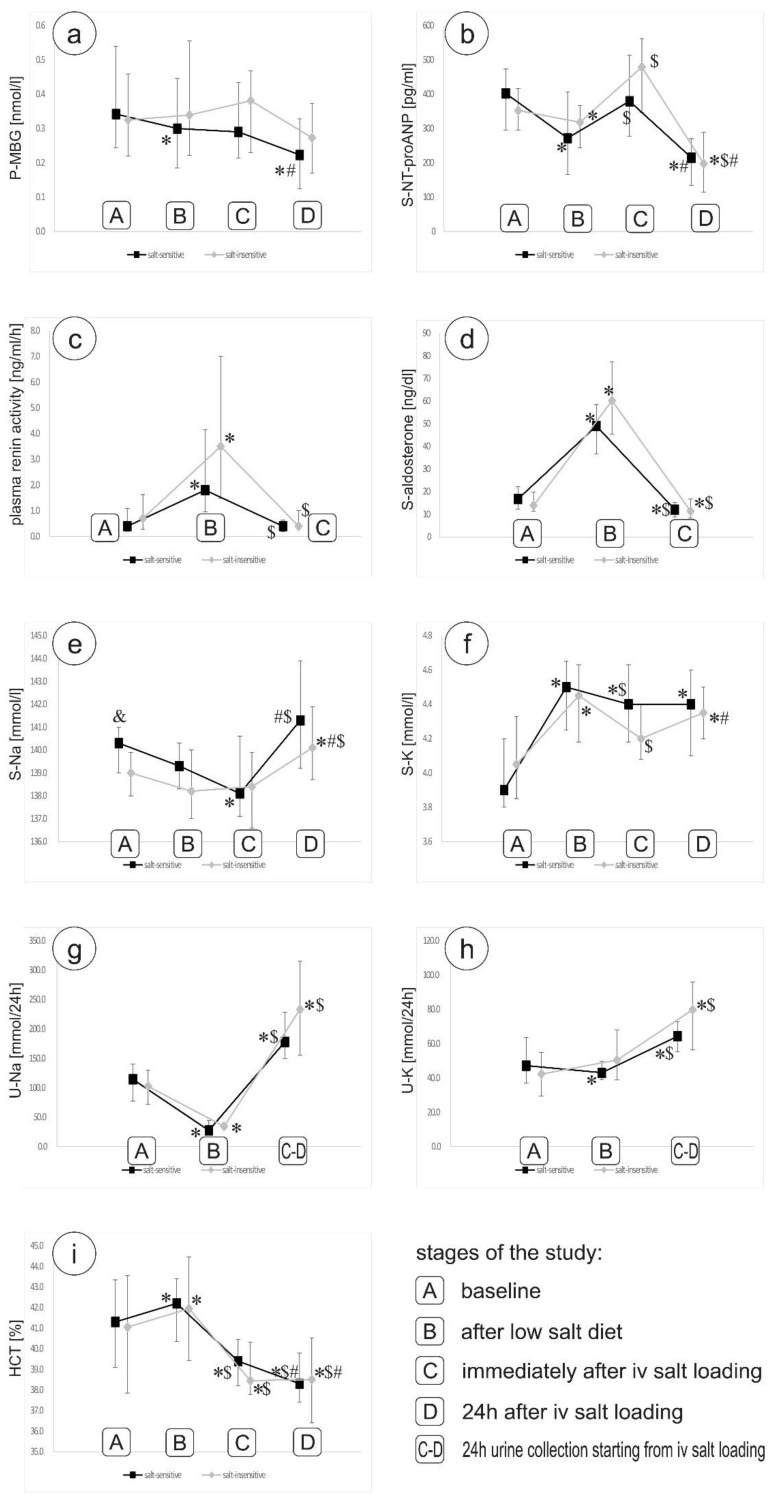
Changes in plasma MBG concentration (**a**) and serum NT-proANP (**b**) concentrations, PRA (**c**), serum aldosterone (**d**), sodium (**e**) and potassium (**f**) concentrations, urinary sodium (**g**) and potassium (**h**) excretion, and hematocrit (**i**), throughout the study. * *p* < 0,05 vs. stage A; $ *p* < 0,05 vs. stage B; # *p* < 0,05 vs. stage C; and & *p* < 0.05 between groups.

**Figure 3 nutrients-13-01502-f003:**
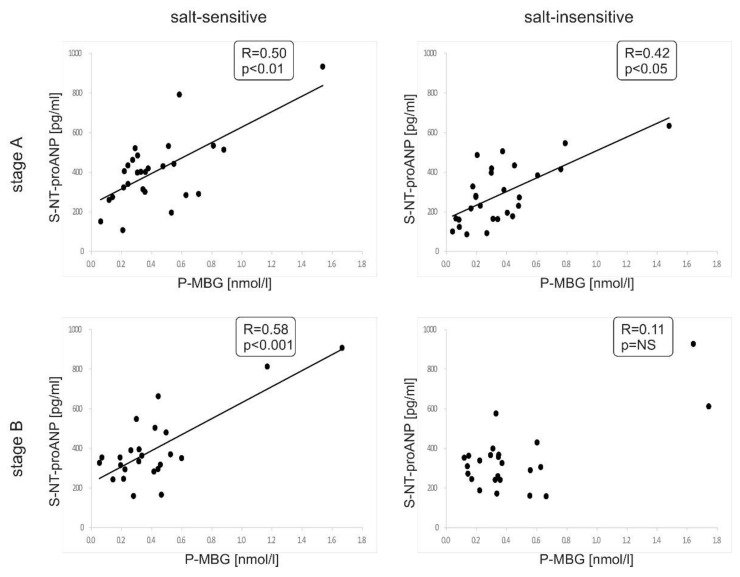
Correlation between plasma MBG and serum NT-pro ANP concentrations at baseline (stage A; upper panels) and after sodium restriction (stage B; lower panels) in salt-sensitive and salt-insensitive groups, respectively.

**Figure 4 nutrients-13-01502-f004:**
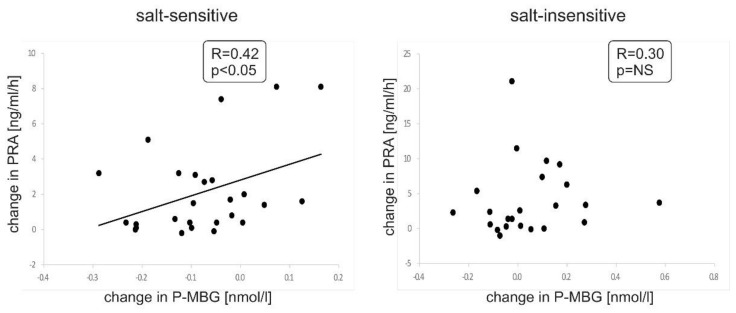
Correlation between changes in plasma MBG (*p*-MBG) from baseline to the end of salt restriction and changes in plasma renin activity (PRA) from baseline to the end of salt restriction in salt-sensitive and salt-insensitive groups, respectively.

**Table 1 nutrients-13-01502-t001:** Basic characteristics of both study groups (salt-sensitive and salt-insensitive patients with primary hypertension).

	Salt-Sensitive Arterial Hypertension *n* = 27	Salt-Insensitive Arterial Hypertension *n* = 24	*p*
**Demographic and clinical characteristics**			
Age at the time of study [yrs]	57.0 (49.0; 59.5)	45.5 (39.3; 60.3)	NS
Sex (female) [*n* (%)]	20 (74.1)	14 (58.3)	NS
BMI [kg/m^2^]	29.4 (27.9; 33.8)	28.2 (26.6; 32.4)	NS
BSA [m^2^]	1.91 (1.84; 2.06)	1.93 (1.82; 2.08)	NS
Overweight [*n* (%)]	8 (29.6)	10 (41.7)	NS
Obesity [*n* (%)]	13 (48.1)	10 (41.7)	NS
Active smokers [*n* (%)]	6 (22.2)	5 (20.8)	NS
Diabetes [*n* (%)]	4 (14.8)	4 (16.7)	NS
Impaired fasting glucose [*n* (%)]	2 (7.4)	0	NS
Coronary heart disease [*n* (%)]	4 (14.8)	3 (12.5)	NS
Previous episode of myocardial infarction [*n* (%)]	1 (3.7)	3 (12.5)	NS
Chronic heart failure [*n* (%)]	1 (3.7)	2 (8.3)	NS
Atherosclerosis [*n* (%)]	5 (18.5)	1 (4.2)	NS
Baseline eGFR [ml/min/1.73 m^2^]	104.0 (95.5; 114.5)	100.0 (94.8; 116.8)	NS
**BP measurements and antihypertensive treatment**			
Baseline SBP [mmHg]	150 (145; 155)	140 (138; 150)	*p* < 0.01
Baseline DBP [mmHg]	90 (90; 100)	85 (79; 90)	*p* < 0.01
Baseline MAP (mmHg]	112 (108; 113)	107 (98; 109)	*p* < 0.01
Use of antihypertensive medication [*n* (%)]	26 (96.3)	24 (100)	NS
Number of antihypertensive drugs at baseline	2 (1; 3)	2 (2; 3)	NS
Change in MAP from baseline to the end of salt restriction [mmHg]	20.0 (16.7; 24.5)	3.3 (0.0; 5.4)	*p* < 0.001
**ABPM parameters at baseline**			
Complete 24-h SBP [mmHg]	132.4 (126.2; 137.1)	126.4 (118.5; 137.4)	NS
Complete 24-h DBP [mmHg]	78.3 (74.2; 84.2)	76.6 (70.7; 83.7)	NS
Daytime SBP [mmHg]	135.1 (130.3; 141.5)	129.9 (124.5; 138.8)	NS
Daytime DBP [mmHg]	80.7 (76.4; 88.2)	78.6 (72.9; 85.1)	NS
Nighttime SBP [mmHg]	120.6 (114.4; 125.4)	110.1 (104.0; 136.1)	NS
Nighttime DBP [mmHg]	71.5 (68.2; 75.0)	68.0 (60.9; 79.0)	NS
Decline in nighttime SBP [mmHg]	10.0 (6.8; 14.0)	12.5 (3.5; 15.1)	NS
Decline in nighttime DBP [mmHg]	10.4 (7.9; 17.9)	12.6 (7.8; 19.1)	NS
**Echocardiographic parameters at baseline**			
LVM [g]	234.0 (217; 279)	245.5 (205.3; 296.0)	NS
LVMI [g/m^2^]	129.0 (110.5; 141.0)	126.5 (109.8; 155.8)	NS
RWT	0.440 (0.415; 0.440)	0.425 (0.400; 0.440)	NS
Left ventricular hypertrophy			
No LVH [*n* (%)]	3 (11.1)	6 (25.0)	NS
Mild [*n* (%)]	4 (14.8)	1 (4.2)	NS
Moderate [*n* (%)]	5 (18.5)	5 (20.8)	NS
Severe [*n* (%)]	15 (55.6)	12 (50.0)	NS
Left ventricular geometric patterns			
Normal geometry [*n* (%)]	2 (7.4)	4 (16.7)	NS
Concentring remodeling [*n* (%)]	1 (3.7)	2 (8.3)	NS
Eccentric hypertrophy [*n* (%)]	6 (22.2)	7 (29.2)	NS
Concentric hypertrophy [*n* (%)]	18 (66.7)	11 (45.8)	NS
**Results of the ophthalmoscopy**			
No visible abnormalities [*n* (%)]	4 (14.8)	9 (37.5)	NS
Hypertensive retinopathy, st. 1 [*n* (%)]	10 (37.0)	10 (41.7)	NS
Hypertensive retinopathy, st. 2 [*n* (%)]	13 (48.2)	5 (20.8)	NS
**Laboratory parameters at baseline**			
MBG [nmol/L]	0.342 (0.243; 0.540)	0.325 (0.220; 0.459)	NS
NT-proANP [pg/mL]	402 (296; 473)	353 (295; 416)	NS
NT-proBNP [pg/mL]	85.0 (26.0; 126.0)	77.5 (37.0; 114.5)	NS
Aldosterone [ng/dl]	16.7 (12.3; 22.2)	13.9 (11.3; 19.7)	NS
PRA [ng/mL/h]	0.40 (0.20; 1.10)	0.70 (0.28; 1.63)	NS
Sodium [mmol/L]	140.3 (139.0; 141.0)	139.0 (138.0; 139.9)	*p* < 0.05
Potassium [mmol/L]	3.90 (3.80; 4.20)	4.05 (3.85; 4.33)	NS
Hematocrit [%]	41.30 (39.10; 43.35)	41.05 (37.85; 43.55)	NS
Urinary sodium excretion [mmol/day]	114.4 (77.3; 140.5)	102.7 (72.4; 130.3)	NS
Urinary potassium excretion [mmol/day]	47.20 (37.10; 63.46)	42.28 (29.51; 54.98)	NS

Data presented as median with interquartile range or numbers and frequencies. BMI, body mass index.; BSA, body surface area; eGFR, estimated glomerular filtration rate; ABPM, ambulatory blood pressure monitoring; SBP, systolic blood pressure; DBP, diastolic blood pressure; LVM, left ventricular mass; LVMI, left ventricular mass index; RWT, relative wall thickness; LVH, left ventricular hypertrophy; MBG, marinobufagenin; NT-proANP, *n*-terminal atrial natriuretic pro-peptide; NT-proBNP, *n*-terminal B-type natriuretic pro-peptide; PRA, plasma renin activity; NS, not significant.

## Data Availability

The data presented in this study are available on request from the corresponding author.

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
