# Peer review of "The Effects of Short-Term Changes in Sodium Intake on Plasma Marinobufagenin Levels in Patients with Primary Salt-Sensitive and Salt-Insensitive Hypertension"

_nutrients, 2021, doi:10.3390/nu13051502_

Round 1

Reviewer 1 Report

The present work is an attempt to combine such problems as hypertension, excessive salt intake and endogenous cardiotonic steroids. The group of patients with primary (essential) hypertension who took part in this study was divided into two subgroups: salt-sensitive and salt-insensitive. On the basis of a large number of parameters, the only significant difference was found between them: the sodium content in plasma was higher in salt-sensitive subjects. In addition, the authors found that in the case of the salt-sensitive subgroup, salt restriction resulted in a decrease in plasma MBG. Surprisingly, acute salt loading resulted in a further decrease in plasma MBG. The authors conclude that "decreased MBG secretion in response to acute salt loading may play an important role in the pathogenesis of salt sensitivity." To be honest, I don't really understand this logic. If I am not mistaken, the authors believe that endogenous MBG acts as an inhibitor of Na,K-ATPase in the kidneys, which causes the physiological effects observed in hypertension (e.g., increased BP value). As an argument, the study of  rat kidney and aorta Na,K-ATPase activity modulation by MBG in the presence of ANP is mentioned. It seems to me that it is worthwhile to extrapolate data from rat to human with great care. The fact is that in rodents the so-called CTS-resistant α1-NKA, and the effects of CTS are often completely different from those in the case of CTS-sensitive α1-NKA (for review, see O.D. Lopina, A.M. Tverskoi, E.A. Klimanova, S.V. Sidorenko, and S.N. Orlov. "Ouabain-induced cell death and survival. Role of a1-Na,K-ATPase-mediated signaling and [Na+]i/[K+]i-dependent gene expression". Frontiers in physiology, 2020). In addition, MBG has a very low affinity for the CTS-sensitive α1-NKA. In my opinion, these facts should be taken into consideration when discussing. As for the study itself and its presentation, I have the following questions and comments:

  1. Did the plasma MBG concentration change over time in the control group?
  2. 1c, d: why are only 3 points presented and where is stage D?
  3. 1h, i: why are only 3 dots represented what does “C-D” mean?
  4. 1: it is necessary to reflect in the legend to the figure what A-D means, since it is inconvenient to refer to the text every time. I think that excessive abbreviations in the axis labels are unnecessary. The graphs themselves need to be redone, now they look carelessly and very difficult to read.
  5. I think that the legends to the rest of the figures should also be redone. Now, in order to understand what the figure is about, you need to refer to the text of the article.
  6. (204-206): Can a 0.05% change in a parameter value be taken as "a progressive increase"?

Author Response

Reviewer 1:

English language and style

( ) Extensive editing of English language and style required
( ) Moderate English changes required
( ) English language and style are fine/minor spell check required
(x) I don't feel qualified to judge about the English language and style

Yes

Can be improved

Must be improved

Not applicable

Does the introduction provide sufficient background and include all relevant references?

( )

( )

(x)

( )

Is the research design appropriate?

( )

(x)

( )

( )

Are the methods adequately described?

( )

(x)

( )

( )

Are the results clearly presented?

( )

( )

(x)

( )

Are the conclusions supported by the results?

( )

( )

(x)

( )

Comments and Suggestions for Authors

The present work is an attempt to combine such problems as hypertension, excessive salt intake and endogenous cardiotonic steroids. The group of patients with primary (essential) hypertension who took part in this study was divided into two subgroups: salt-sensitive and salt-insensitive. On the basis of a large number of parameters, the only significant difference was found between them: the sodium content in plasma was higher in salt-sensitive subjects. In addition, the authors found that in the case of the salt-sensitive subgroup, salt restriction resulted in a decrease in plasma MBG. Surprisingly, acute salt loading resulted in a further decrease in plasma MBG. The authors conclude that "decreased MBG secretion in response to acute salt loading may play an important role in the pathogenesis of salt sensitivity." To be honest, I don't really understand this logic. If I am not mistaken, the authors believe that endogenous MBG acts as an inhibitor of Na,K-ATPase in the kidneys, which causes the physiological effects observed in hypertension (e.g., increased BP value). As an argument, the study of  rat kidney and aorta Na,K-ATPase activity modulation by MBG in the presence of ANP is mentioned. It seems to me that it is worthwhile to extrapolate data from rat to human with great care. The fact is that in rodents the so-called CTS-resistant α1-NKA, and the effects of CTS are often completely different from those in the case of CTS-sensitive α1-NKA (for review, see O.D. Lopina, A.M. Tverskoi, E.A. Klimanova, S.V. Sidorenko, and S.N. Orlov. "Ouabain-induced cell death and survival. Role of a1-Na,K-ATPase-mediated signaling and [Na+]i/[K+]i-dependent gene expression". Frontiers in physiology, 2020). In addition, MBG has a very low affinity for the CTS-sensitive α1-NKA. In my opinion, these facts should be taken into consideration when discussing.

Answer:

Manuscript was supplemented with information in the introduction.

The pathogenesis of salt sensitivity of blood pressure is not entirely understood. It was postulated that increased MBG secretion after high sodium load is an attempt to compensate for impaired pressure-natriuresis in the animal model of salt-sensitivity. As mentioned above, there are significant differences between species (rodents vs. humans) in terms of the sensitivity of particular NKA isoforms to different CTS. So far, no studies have been conducted to determine the effects of short-term changes in sodium intake on plasma MBG levels in patients with primary hypertension depending on salt sensitivity. In the present study we did not find significant differences in the plasma MBG concentration between salt-sensitive and salt-insensitive patients, both after sodium dietary restriction and after intravenous sodium load. However, crucial changes in plasma MBG concentration were noted at subsequent stages of observation only in the salt-sensitive group. There was no significant increase in the plasma MBG level in response to acute sodium chloride load in our cohort, in contrast to again cited experimental studies conducted in rats. Only one study performed in 34 patients with resistant hypertension showed that acute salt loading (60 min via intravenous infusion of 1 L of 0.9% NaCl) resulted in the increase of plasma MBG concentration immediately after the salt loading, but all patients in this study received a thiazide diuretic and salt loading resulted in a substantial BP rise. We concluded, that decreased MBG secretion in response to acute salt loading may play an important role in the pathogenesis of salt sensitivity. One can assume, that decreased MBG secretion after sodium load may lead to sodium and water retention in salt-sensitive patients. Additionally, this impaired sodium excretion promotes other mechanism involved in the pressure induced mechanism of natriuresis. It is a hypothesis, which should be further investigated in humans.

  1. Did the plasma MBG concentration change over time in the control group?

Answer:

Plasma MBG concentration in salt-sensitive individuals tended to change during observation, as presented in Fig. 1 . We have noted a slightly increase in plasma MBG directly after acute sodium loading, however the difference between subsequent stages of this study did not reach statistical significance in the salt-insensitive group.

  1. 1c, d: why are only 3 points presented and where is stage D?

Answer:

Serum concentrations of aldosterone and plasma renin activity (PRA) were assessed only at stage 1-3 of this study. This information was corrected in point 2. Materials and Methods.

  1. 1h, i: why are only 3 dots represented what does “C-D” mean?

Answer:

Urinary sodium and potassium excretion was measured in 24-hour urine collection at the beginning (stage A), during dietary sodium restriction (stage B) and during acute salt loading (stage C-D). Stage C-D means- 24 hour urine collection that began immediately after acute intravenous sodium chloride load (stage C) and ended after 24 hours (stage D). It was corrected in point 2. Materials and Methods, too.

  1. 1: it is necessary to reflect in the legend to the figure what A-D means, since it is inconvenient to refer to the text every time. I think that excessive abbreviations in the axis labels are unnecessary. The graphs themselves need to be redone, now they look carelessly and very difficult to read.
  2. I think that the legends to the rest of the figures should also be redone. Now, in order to understand what the figure is about, you need to refer to the text of the article.

Answer:

The figures and the legends have been corrected.

  1. (204-206): Can a 0.05% change in a parameter value be taken as "a progressive increase"?

Answer:

 We agree with the reviewer, a 0.05% change in a parameter value should not be taken as "a progressive increase"- the sentence has been corrected.

Reviewer 2 Report

The main aim of this study was to determine the effects of short-term changes in sodium intake on plasma MBG levels in 51 primary hypertension patients, stratified according to salt-sensitivity. This prospective study was conducted for 4 days, and four observation points were evaluated: at baseline (stage A), after a 3-days low sodium diet (≤20 mmol Na/day) (stage B), after acute intravenous sodium chloride load (310 mmol sodium, 2L of 0.9% sodium chloride infused over 4 h) (stage C) and 24 hours after the salt loading (stage D). In the salt-sensitive group, plasma MBG decreased with dietary sodium restriction, and with further reductions 24 hours after salt loading (stage D). No significant effect of short-term changes in sodium intake on plasma MBG levels was observed in the salt-insensitive group.

I have the following comments:

  1. Introduction

1.1          The following sentence needs to be edited: “Marinobufagenin, by renotubular NKA inhibition initiates natriuresis and induces vascular smooth muscle cells (VSMC) contraction [20,21]”. Currently it sounds as if MBG promotes vasoconstriction via inhibition of renal NKA. It should be clearly stated that it is the inhibition of VSMC NKA that promotes vasoconstriction.

  1. Materials and methods

2.1          This study included 51 patients that were diagnosed as being hypertensive based on attended office BP measurements (≥140/90 mmHg) or antihypertensive medication use. There are some important questions regarding the hypertensive status of participants in this study.

From what I can assume, BP was measured only once on the left arm of participants while they were in a seated position. Why was BP not measured 2-3 times, as hypertension based on office BP is diagnosed after multiple measurements? Why were baseline ABPM measurements not considered to diagnose hypertension?

From the basic characteristics, only the salt-sensitive group had ABPM blood pressures that fulfilled the criteria for hypertension based on the ISH guidelines (24h average ≥ 130 and or ≥ 80 mmHg; daytime average ≥ 135 and or ≥ 85 and night-time average ≥ 120 and or ≥ 70 mmHg). ABPM blood pressures for salt-insensitive patients were lower than all of the aforementioned cut points (24h average 126/77 mmHg; daytime average 130/79 mmHg and night-time average 110/68 mmHg). It seems possible that the white coat effect may have played a role in this subgroup if participants had a high office BP but normotensive ABPM. What proportion of this group made use of antihypertensive medication to justify their inclusion into this study as patients with primary hypertension? Please include this important information into Table 1. Currently, it seems as if the salt-sensitive subgroup was hypertensive, but the salt-insensitive subgroup was normotensive based on ABPM.

  1. Results and discussion

3.1          Patients were classified as salt-sensitive when MAP on a low-sodium diet was at least 10 mmHg lower than the MAP value after sodium load. It is possible that the white coat effect could have had an influence on the MAP that was calculated at each time point. In the introduction the authors highlight significant characteristics that are associated with salt sensitivity, including insufficiently suppressed RAAS and decreased atrial natriuretic peptide (ANP). However, the salt-sensitive group in this study population exhibited similar plasma renin, serum aldosterone, NT-proANP, urinary sodium and potassium levels compared to the salt-insensitive group irrespective of sodium intake. Also, similar responses in the levels of these biomarkers were observed with the respective changes in sodium load. The authors cite one other study that did not find differences in PRA or aldosterone concentrations between salt-sensitive and salt resistant individuals. Still, the authors should further elaborate on why they do not observe any significant differences in the concentrations of the aforementioned biomarkers between salt-sensitive and non-salt-sensitive patients in their cohort.

3.2          Please indicate the median change in MAP for the salt-insensitive group, and did it differ significantly from the median change in MAP from the salt-sensitive group? How many patients in the salt-insensitive group where salt resistant (<5 mmHg decrease or an increase in BP) and how many had an indeterminate BP change of -5 to -9 mmHg)?

3.3       In section 2.5 the authors report the correlations between MBG and other parameters, but MBG concentration and change in MBG concentration is used interchangeably as a variable. The authors report a correlation of NT-proANP concentration as well as aldosterone concentration with plasma MBG concentrations. But report the correlation between change in MBG concentration and change in PRA. Why not report the results for the correlation of the change in NT-proANP or change in aldosterone with change in MBG?

3.4          Did the authors consider looking at the correlation between plasma MBG and potassium (serum and urinary)? It would be interesting to see the relationship in this cohort. The increase in potassium after NaCl intervention, as observed in this study, may also play a role. Fedorova et al. have previously discussed the possible role of increased potassium on plasma MBG levels (Journal of Hypertension 2015; 33: 534-541).

3.5        If possible, the authors should also include figures to demonstrate the change in SBP, DBP and LVM at stages A to D.

3.6        So there were no correlations between MBG and SBP or DBP during sodium restriction or after NaCl infusion?

      4.       Conclusion

The authors conclude that the hypotensive effect of reduced dietary sodium may be due to a decline in adrenal MBG synthesis and release in sodium-sensitive subjects. Referring to the hypotensive effect of MBG may be a to strong statement that is not fully supported by a single correlation between baseline MBG and 24hr DBP. The authors fail to demonstrate if lower plasma MBG levels are associated with a decrease in DBP or SBP during any of the stages of sodium intervention. Therefor an attenuated pressor response associated with MBG levels is not demonstrated. I would change this sentence. It is also possible that the lower plasma MBG, which correlates with lower 24hr DBP, may be due to increased urinary MBG excretion and not necessarily decreased MBG synthesis or secretion.

Round 2

Reviewer 1 Report

I thank the authors for their attention to my comments. In my opinion, the manuscript has become much better and clearer. But I would still recommend to make the axis labels in a larger font.